# Knowledge and practices regarding prostate cancer screening in Spanish men: The importance of personal and clinical characteristics (PROSHADE study)

Lucy A. Parker[1,2], Juan-Pablo Caballero-Romeu[3], Elisa Chilet-Rosell[1,2], Ildefonso Hernandez-Aguado[1,2], Luis Gómez-Pérez[4], Pablo Alonso-Coello[5], Ana Cebrián[6], Maite López-Garrigós[2,7], Irene Moral-Pélaez[8], Elena Ronda[2,9], Mercedes Gilabert[10], Carlos Canelo-Aybar[2,5], Ignacio Párraga-Martínez[11], Mª del Campo-Giménez[12], Blanca Lumbreras[1,2]*

1 Department of Public Health, History of Science and Gynecology, Miguel Hernandez University, San Juan de Alicante, Spain, 2 CIBER of Epidemiology and Public Health, CIBERESP, Madrid, Spain, 3 Department of Urology, Dr. Balmis General University Hospital; Alicante Institute for Health and Biomedical Research (ISABIAL), Alicante, Spain, 4 Department of Urology, University General Hospital of Elche, Elche, Spain, 5 Iberoamerican Cochrane Centre - Department of Clinical Epidemiology and Public Health, Biomedical Research Institute Sant Pau, Barcelona, Spain, 6 Cartagena Casco Healthcare Centre, Cartagena, Spain, 7 Clinical Laboratory, Hospital Universitario de San Juan, San Juan de Alicante, Spain, 8 EAP Sardenya, Barcelona. Institut de Recerca Sant Pau, Barcelona, Spain, 9 Public Health Research group, Alicante University, San Vicente del Raspeig, Spain, 10 Department of Health Psychology, Miguel Hernandez University, Elche, Spain, 11 Health Care Center Zone VIII, Servicio de Salud Castilla-La Mancha, Albacete, Spain, 12 Integrated Care Management of Albacete. Health Service of Castilla-La Mancha, Spain

* blumbreras@goumh.umh.es

## Abstract

### Introduction

Patients' decisions on prostate cancer (PCa) opportunistic screening may vary. This study aimed to assess how demographic and health-related characteristics may influence knowledge and decisions regarding PCa screening.

### Methods

A cross-sectional survey was conducted among men aged over 40, randomly sampled from the Spanish population, 2022. The survey underwent development and content validation using a modified Delphi method and was administered via telephone. Binomial logistic regression was used to explore the relationship between respondents' characteristics and participants' knowledge and practices concerning PCa and the PSA test.

### Results

Out of 1,334 men, 1,067 (80%) respondents were interviewed with a mean age of 58.6 years (sd 11.9). Most had secondary or university studies (787, 73.8%) and 61 (5.7%) self-reported their health status as bad or very bad. Most of the respondents (1,018, 95.4%) had knowledge regarding PCa with nearly 70% expressed significant concern about its potential

**Data Availability Statement:** All relevant data are within the manuscript and its Supporting information files.

**Funding:** Research funded by the Instituto de Salud Carlos III, code PI20/01334, Principal Investigator Dr. Blanca Lumbreras Lacarra, co-financed with FEDER funds from the European Union "A way of doing Europe". The research is independent, however, and the views expressed in this article are solely those of the authors. The funders had no role in study design, data collection and analysis, decision to publish, or preparation of the manuscript.

**Competing interests:** The authors have declared that no competing interests exist.

development (720, 70.8%), particularly among those under 64 years (p = 0.001). Out of 847 respondents, 573 (67.7%) reported that they have knowledge regarding the PSA test: 374 (65.4%) reported receiving information from a clinicians, 324 (86.6%) information about the benefits of the test and 189 (49,5%) about its risks, with differences based on educational background. In a multivariable analysis (adjusted for age, educational level and previous prostate problems), respondents with higher levels of education were more likely to have higher knowledge regarding the PSA test (OR 1.75, 95%CI 1.24–2.50, p<0.001).

## Conclusions

Although most of the patients reported to have knowledge regarding PCa, half of the interviewed men reported knowledge about PSA test. Differences in knowledge prostate cancer screening and undesirable consequences highlight the need to develop and provide tailored information for patients.

## Introduction

Population-based screening for prostate cancer (PCa) using prostate-specific antigen (PSA) test is controversial because it is often associated with overdiagnosis [1]. In addition, previous studies carried out in clinical practice have shown a high rate of false positive results [2]. Thus, although PSA screening has been associated with a small absolute reduction in disease-specific mortality [3], it is unclear whether its benefits outweigh its potential harms. This uncertainty is reflected in significant variability in screening practices worldwide [4]. However, consensus does exist on the need to incorporate a strategy of shared-decision making (SDM) when PCa screening is considered. The U.S. Preventive Services Task Force (USPSTF) updated its recommendations in 2017. They stated that opportunistic screening may be useful for men aged 55–69 years, but the decision to screen should be made by each patient together with a physician after the patient has understood the benefits and risks of screening [5]. The European Association of Urology (EAU) [6] updated their recommendations in 2015 and more recently in 2021. They stated that clinicians should offer an individualized early detection strategy to inform patients aged over 50 years old with a good functional status and a life expectancy of at least 10–15 years old, and to those at higher risk of PCa.

Different approaches have been developed to provide clinicians with SDM resources. The Agency for Healthcare Research and Quality (AHRQ) SHARE Approach includes 5 different steps to achieve SDM, one of which is to help patients explore and compare treatment options. However, patient's lack of knowledge about PCa and about medical recommendations on early detection of PCa may be a barrier to making an informed decision about PSA screening [7]. With any SDM resource, for clinicians to help patients choose the right option, it is essential that they first assess the patients' knowledge about all the available options [8].

Furthermore, they should be aware that patients' decisions related to their medical options can vary. For example, a patient may choose different clinical options at different points during his life, according to a change in his personal and clinical characteristics [9].

Evidence suggests that patients have a low level of knowledge about screening tests and particularly PSA. In addition, published data on knowledge of PSA and screening tests in general are limited, and recently published data refer mainly to disadvantaged and high-risk populations [10–12].

However, little evidence exists about the clinical and sociodemographic factors that can influence patients' knowledge and practice regarding the use of PSA. Although a previous

systematic review focused on characteristics associated with knowledge about PCa and PCa screening in Black African and Black Caribbean men [13] and also in a sample of Italian men in 2012 [14], no studies have carried out such research since the updating of available guidelines. For clinicians, understanding how sociodemographic and clinical factors can influence patients' knowledge and usual practice is critical for the development and implementation of interventions to discuss the benefits and limitations of the PSA test with them according to available guidelines.

The aim of this study was to evaluate how the personal and clinical factors influence men's knowledge and practice about PCa and the use of the PSA test for the opportunistic screening of PCa.

## Material and methods

The present study, conducted in 2022 in the Valencia Community, Southeast Spain, aimed to assess the population's knowledge, opinions, and practices regarding PCa and the PSA test in opportunistic PCa screening. Protocol of the PROSHADE study has been previously published [15]. The manuscript follows the STROBE [16] recommendation (S1 Table).

### Design

Cross-sectional study based on a population survey.

### Setting

The Valencia Community has a population of men in the relevant range of age (older than 40 years) of 1,406,419 inhabitants, divided into three provinces: Alicante, Castellón, and Valencia, with populations of 528,561, 165,953, and 711,905 inhabitants, respectively.

### Subjects

Men aged 40 or older without a prior PCa diagnosis were randomly selected by telephone using the Computer-Assisted Telephone Interviewing (CATI) platform. The selection was stratified according to province (Alicante, Castellón, Valencia), municipality size (> 100.000 habitans, 50–100.000 habitans, 20–50.0000 habitans, 5–20.000 habitans, <5.000 habitans) and age (3 intervals: 40–49 years old, 50–64 years old, >64 years old) of the population of men aged 40 and over without a diagnosis of PCa. Subjects were selected randomly with proportional affixation to the reference population in each stratum.

### Sample size

To achieve a precision of 3% with a 95% confidence interval, a conservative estimate suggested a need for at least 1,066 men. This estimation accounted for the possibility that 50% of men might be unaware of the PSA test benefit/risk. This precision allows for subgroup analysis.

### Questionnaire

A questionnaire was designed to analyze the general male population' knowledge, opinions and practice about PCa opportunistic screening with the PSA test. A modified Delphi method was used for the questionnaire's development and content validation.

In the initial phase, a group of experts were invited to participate in the procedure: Family and Community Medicine Clinicians (4), Clinical Epidemiologists (4), Urologists (2), Clinical laboratory physician (1) and Psychologist (1) with research experience and the ability to provide in-depth opinions by connecting scientific evidence with practical experience. The experts

were contacted via email. The anonymity of the experts was maintained throughout to prevent biases.

In the exploratory phase, the expert panel received an email invitation explaining the study's objectives and the Delphi method. The coordinating group drafted an initial version based on literature review. Virtual voting by the expert panel followed, using a Likert scale (where 1 meant unimportant and 5 very important) to score each category or item's relevance for inclusion in the final questionnaire. Experts could also provide "observations" for suggestions or corrections. A second round of virtual voting occurred, considering modifications proposed by experts in the first round. A 14-day response period was given after each round.

The final phase included a pilot study, where the definitive questionnaire was submitted to the expert panel for approval and after which, the questionnaire was given to 15 men older than 40 years old to test comprehension and measure the completion time. There were no significant changes in the questionnaire in this pilot study.

## Variables included

The variables included in the questionnaire were: age (years), educational level (primary, secondary/university), self perceived health status (very bad/bad, normal, very good/good), previous prostate problems (self or relatives), previous cancer (excluded PCa, self or relatives).

## Procedure

The survey was conducted by telephone using the Computer-Assisted Telephone Interviewing (CATI) platform, with a maximum questionnaire duration of 12 minutes. Interviewers were selected based on their fit with the study profile and received prior training.

Respondents gave their informed consent orally to the interviewer.

## Statistical analysis

The data collected in the study was coded and entered into IBM SPSS Statistics 27.0 (IBM Corp., Armonk, NY, USA).

Questionnaire analysis: Statistical analysis involved assessing item scores for selection, using the mean and Aiken's V test. Content validity was analyzed through item averages, Aiken's V test, and qualitative expert assessments to adjust questionnaire categories. Criteria for item selection were a mean score above 3.5 and Aiken's V test result greater than or equal to 0.70. Items with a lower 95% confidence interval limit below 0.70 were included if the mean exceeded 3.5 and the median was 4 or higher. Reliability was evaluated using Cronbach's alpha coefficient. We collected the expert's evaluations using Google Forms, and analyzed the data with IBM SPSS Statistics v27 for Windows.

We assume that missing values occurred at random (there were no differences between those cases who answered a particular question and those who did not) and thus, we applied the complete case to deal with missing data (we omitted those cases with missing data and analyze the remaining data).

The descriptive statistics for categorical variables were expressed in numbers and percentages. The association between respondents' factors (age, educational level, self-reported health status, previous prostate cancer and previous cancer) and levels of knowledge and practice towards PCa and the PSA test were tested using the Chi-square test, and a p-value <5% was considered statistically significant. Binomial logistic regression was used to test the likelihood of: a) having knowledge regarding PCa, b) having knowledge regarding the PSA test, c) having received information by the physician regarding PSA test, and d) having received information about the risks regarding the PSA test, according to respondents' characteristics. We presented

adjusted odd ratios with their corresponding 95% confidence intervals and a p-value <5% was considered statistically significant.

## Ethic statement

Institutional Review Board Statement: CEIC Sant Joan d'Alacant (20/041) on 8th January 2021.

Informed Consent Statement: All participants will give oral informed consent prior to entry to the study by a member of the study team and will be made aware that participation is strictly voluntary.

## Results

### Socio-demographic characteristics of the respondents

Out of 1,334 contacted men, 1,067 (80%) respondents were interviewed (refusal rate of 20%, with no differences by age group, province or municipality) (Fig 1), with a mean age of 58.6 years (sd 11.9). Most had secondary or university studies (787, 73.8%) and 61 (5.7%) self-reported their health status as bad or very bad. Less than 30% of the respondents reported pervious prostate problems (self or relatives) and 425 (40.2%) reported a previous cancer (self or relatives) (not presented in tables).

### Respondents' knowledge regarding prostate cancer

The majority of respondents (1,018, 95.4%) indicated that they had knowledge regarding PCa. Respondents aged 40–49 years (322, 97.9%) and those aged 50–64 years (392, 96.3%) were more likely to report that they have knowledge regarding PCa than those over 64 years (304, 91.8%), p = 0.001. In addition, respondents with secondary or university studies (770, 97.8%) were more like to report that they have knowledge regarding PCa than those with primary education (236, 88.1%), p<0.001. In a multivariable analysis adjusted by age, it was observed that respondents with secondary or university education were significantly more likely to report that they have knowledge regarding PCa compared to those with primary education (aOR 5.23, CI95% 2,84–9.76, p<0.001) (not presented in tables).

While most respondents who reported that they had knowledge about PCa believed that men between 50–70 years had a higher risk of developing PCa (664, 70%), a notable percentage (232, 24.5%) thought that men under 50 years were the most at risk (Table 1). In addition, there were differences in response depending on the age of the respondents: those aged 40–49 years were more likely to think that men under 50 years were the most at risk group, compared to those aged 50–64 years and those over 64 years, p<0.001. However, most of the respondents believed that the risk of developing PCa increased with age (877, 89.6%%).

Nearly 70% of the respondents expressed significant concern about the possibility of developing PCa (720, 70.8%), mainly those aged 40–49 years (238, 73.9%) and those aged 50–64 (289, 73.9%) in comparison with those over 64 years (193, 63.5%), p = 0.004. Respondents who reported having a bad or very bad state of health were less likely to harbor such concerns (33, 60%) in comparison with those in a normal state of health (323, 75.1%) or in a very good or good state of health (363, 68.5%, p = 0.015).

### Respondents' knowledge regarding the PSA test

Out of 1,067 respondents, 847 (79.4%) answered questions related to knowledge regarding the PSA test. There were no differences in sociodemographic characteristics between those respondents who answered these questions and those who did not.

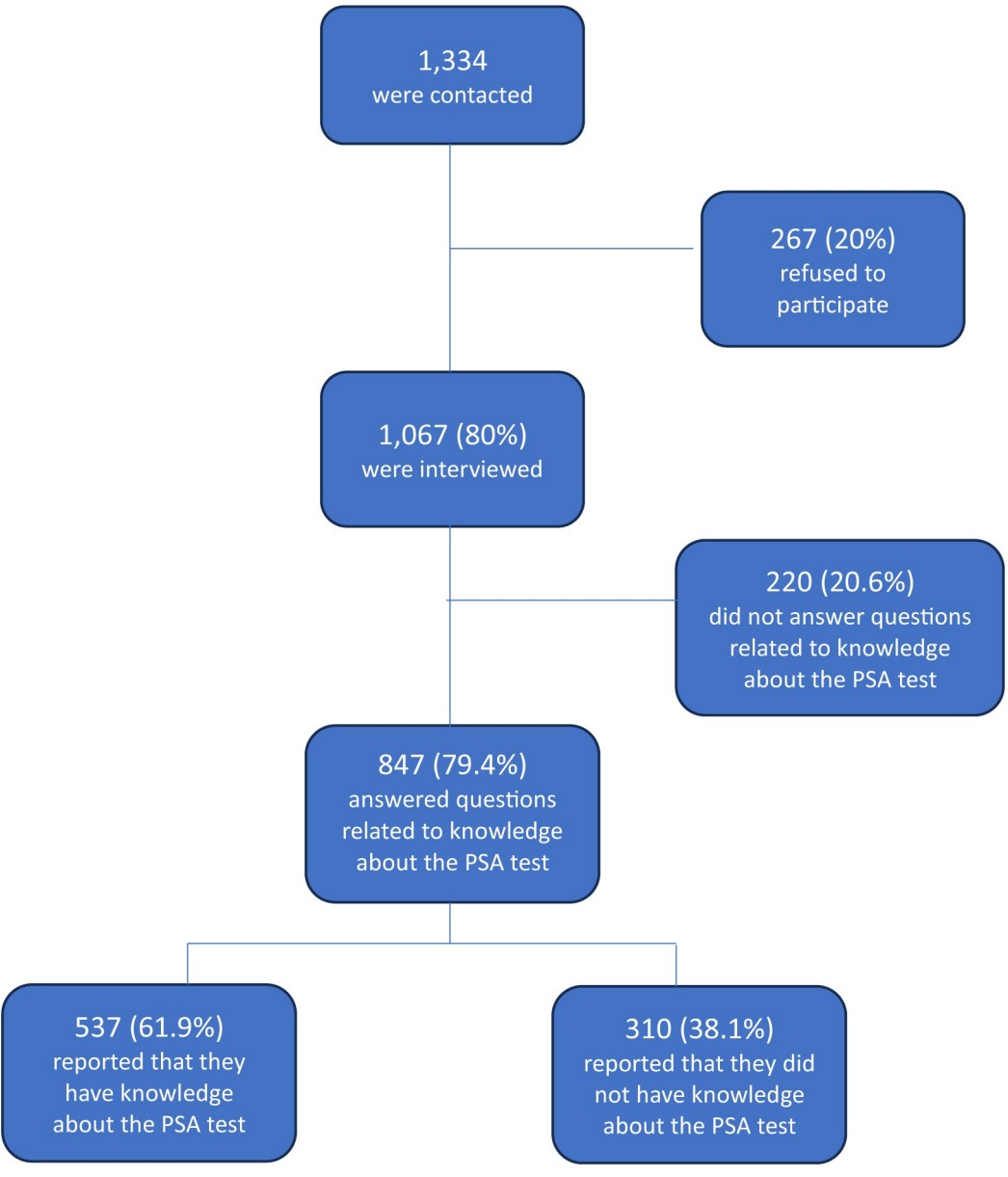

**Fig 1. Flow diagram of respondents included in each section of the study.**

Out of 847 respondents, 573 (67.7%) reported that they have knowledge regarding the PSA test. Respondents aged 50–64 years (249/353, 70.5%) and those over 64 years (190/251, 75.7%) were more likely to report that they have knowledge regarding the PSA test than those aged 40–49 years (134/243, 55.1%), p<0.001. Respondents with secondary or university studies (446/645, 69.1%) were more likely to report having knowledge regarding the PSA test than those with primary education (117/192, 60.9%), p = 0.033. Respondents with previous prostate problems (self or relatives) (179/226, 79.2%) were more likely to report having knowledge regarding the PSA test than those who did not (392/619, 63.3%), p<0.001.

In a multivariable analysis (adjusted by age, educational level and previous prostate problems), respondents over 64 years and those aged 50–64 were more likely to report having

**Table 1. Description of respondents' knowledge and attitudes regarding prostate cancer.** (only those respondents who had knowledge regarding to PCa).

| Variables (n, %) | Total | At what ages do you think men are most at risk of developing PCa? | | | | Does PCa risk increase with age? | | | Do you worry about developing PCa? | | |
|---|---|---|---|---|---|---|---|---|---|---|---|
| | | <50 | 50–70 | >70 | P value | Little/none | Quite a lot/ a lot | P value | Little/none | Quite a lot/ a lot | P value |
| **Age (years)** | | | | | <0.001 | | | <0.001 | | | 0.004 |
| 40–49 | 322 | 113 (36.5) | 185 (59.7) | 12 (3.9) | | 18 (5.8) | 293 (94.2) | | 84 (26.1) | 238 (73.9) | |
| 50–64 | 392 | 85 (22.6) | 272 (72.3) | 19 (5.1) | | 28 (7.4) | 349 (92.6) | | 102 (26.1) | 289 (73.9) | |
| >64 | 304 | 34 (13) | 207 (79) | 21 (8) | | 44 (15.8) | 235 (84.2) | | 111 (36.5) | 193 (63.5) | |
| **Educational level** | | | | | 0.938 | | | 0.162 | | | 0.366 |
| • Primary | 236 | 53 (25.2) | 145 (69) | 12 (5.7) | | 26 (11.8) | 194 (88.2) | | 63 (26.8) | 172 (73.2) | |
| • Secondary/University | 770 | 176 (24.2) | 512 (70.3) | 40 (5.5) | | 64 (8.7) | 673 (91.3) | | 230 (29.9) | 540 (70.1) | |
| **Self-reported health status** | | | | | 0.218 | | | 0.658 | | | 0.015 |
| • Very bad/bad | 55 | 9 (17) | 41 (77.4) | 3 (5.7) | | 6 (11.3) | 47 (88.7) | | 22 (40) | 33 (60) | |
| • Normal | 431 | 100 (24.4) | 294 (71.7) | 16 (3.9) | | 34 (8.4) | 371 (91.6) | | 107 (24.9) | 323 (75.1) | |
| • Very good/good | 530 | 123 (25.4) | 328 (67.8) | 33 (6.8) | | 50 (9.9) | 457 (90.1) | | 167 (31.5) | 363 (68.5) | |
| **Previous prostate problems (self or relatives)** | | | | | 0.122 | | | 0.120 | | | 0.115 |
| • No | 749 | 167 (24.1) | 493 (71.2) | 32 (4.6) | | 72 (10.2) | 634 (89.8) | | 229 (30.6) | 520 (69.4) | |
| • Yes | 268 | 65 (25.5) | 170 (66.7) | 20 (7.8) | | 18 (6.9) | 242 (93.1) | | 68 (25.5) | 199 (74.5) | |
| **Previous cancer (self or relatives)** | | | | | 0.390 | | | 0.571 | | | 0.738 |
| • No | 603 | 139 (25) | 392 (70.4) | 26 (6.7) | | 51 (8.9) | 521 (91.1) | | 174 (28.9) | 429 (71.1) | |
| • Yes | 410 | 93 (24.1) | 267 (69.2) | 26 (6.5) | | 39 (10) | 351 (90) | | 122 (29.9) | 287 (70.2) | |
| **Total** | **1,018** | **232 (24.5)** | **664 (70)** | **52 (5.5)** | | **90 (10.4)** | **877 (89.6)** | | **297 (29.2)** | **720 (70.8)** | |

PCa: Prostate cancer

knowledge regarding PSA than those aged 40–49 years (OR 2.86, 95%CI 1.92–4.27, p<0.001, and OR 2.13, 95%CI 1.49–3.03, p<0.001). Respondents with secondary or university studies were more likely to report having knowledge regarding PSA than those with primary education (OR 1.75, 95%CI 1.24–2.50, p<0.001). (not presented in tables).

**Questions related to knowledge about the PSA test characteristics.** Respondents aged 40–49 years were more likely to believe that the appropriate age for undergoing a PSA test was <50 years (57, 42.5%), in contrast to those aged 50–64 years (82, 32.9%) and those over 64 years (52, 27.4%) (p<0.001). Most respondents felt that the PSA test was beneficial before the onset of symptoms (524, 96.5%). More than half of the respondents (60%) thought that a PSA test decreased the risk of developing PCa with this belief being more prevalent in those with primary education (72, 69.9%) compared to those with secondary/university studies (220, 53.3%), p = 0.002. Lastly, nearly all of the respondents (96.4%) believed that asymptomatic men over 50 years should undergo screening tests (Table 2).

**Table 2. Description of respondents' knowledge about characteristics related to the PSA test.** (only those respondents who stated that they had knowledge regarding to the PSA test).

| Variables (n, %) | Total | Age PSA test | | | | Is the PSA test useful before symptoms appear? | | | Does having a PSA test decrease the risk of developing PCa? | | | Should asymptomatic men >50 be screened with the PSA test for early detection of PCa | | |
|---|---|---|---|---|---|---|---|---|---|---|---|---|---|---|
| | | <50 | 50–70 | >70 | P value | No | Yes | P value | No | Yes | P value | No | Yes | P value |
| **Age (years)** | | | | | 0.026 | | | 0.036 | | | 0.885 | | | 0.250 |
| 40–49 | 134 | 57 (42.5) | 77 (57.5) | 0 | | 1 (0.8) | 125 (99.2) | | 53 (43.1) | 186 (65.3) | | 3 (2.3) | 129 (97.7) | |
| 50–64 | 249 | 82 (32.9) | 165 (66.3) | 2 (0.8) | | 7 (2.9) | 231 (97.1) | | 98 (42.6) | 132 (57.4) | | 7 (2.9) | 235 (97.1) | |
| >64 | 190 | 52 (27.4) | 134 (70.5) | 4 (2.1) | | 11 (6.1) | 168 (93.9) | | 77 (45) | 94 (55) | | 10 (5.4) | 175 (94.6) | |
| **Educational level** | | | | | 0.634 | | | 0.278 | | | 0.002 | | | 0.244 |
| • Primary | 117 | 41 (35) | 74 (63.2) | 2 (1.7) | | 2 (1.8) | 107 (98.2) | | 31 (30.1) | 72 (69.9) | | 2 (1.8) | 110 (98.2) | |
| • Secondary/University | 446 | 145 (32.5) | 297 (66.6) | 4 (0.9) | | 17 (4) | 409 (96) | | 193 (46.7) | 220 (53.3) | | 18 (4.1) | 422 (95.9) | |
| **Self-reported health status** | | | | | 0.146 | | | 0.665 | | | 0.319 | | | 0.196 |
| • Very bad/bad | 24 | 10 (41.7) | 13 (54.2) | 1 (4.2) | | 1 (4.2) | 23 (95.8) | | 8 (36.4) | 14 (63.6) | | 0 | 24 (100) | |
| • Normal | 241 | 73 (30.3) | 164 (68) | 4 (1.7) | | 6 (2.7) | 220 (97.3) | | 89 (40.5) | 131 (59.5) | | 12 (5.1) | 222 (94.9) | |
| • Very good/good | 308 | 108 (35.1) | 199 (64.6) | 1 (0.3) | | 12 (4.1) | 281 (95.9) | | 131 (46.5) | 151 (53.5) | | 8 (2.7) | 293 (97.3) | |
| **Previous prostate problems (self or relatives)** | | | | | 0.132 | | | 0.609 | | | 0.360 | | | 0.921 |
| • No | 392 | 120 (30.6) | 268 (68.4) | 4 (1) | | 12 (3.2) | 360 (96.8) | | 151 (42.1) | 208 (57.9) | | 14 (3.6) | 371 (96.4) | |
| • Yes | 179 | 70 (39.1) | 107 (59.8) | 2 (1.1) | | 7 (4.1) | 164 (95.9) | | 76 (46.3) | 88 (53.7) | | 6 (3.5) | 167 (96.5) | |
| **Previous cancer (self or relatives)** | | | | | 0.932 | | | 0.416 | | | 0.848 | | | 0.285 |
| • No | 324 | 107 (33) | 214 (66) | 3 (0.9) | | 9 (2.3) | 297 (97.1) | | 129 (43.9) | 165 (56.1) | | 9 (2.8) | 307 (97.2) | |
| • Yes | 247 | 83 (33.6) | 161 (65.2) | 3 (1.2) | | 10 (4.2) | 226 (95.8) | | 99 (43) | 131 (57) | | 11 (4.5) | 231 (95.5) | |
| **Total** | **573** | **191 (33.4)** | **376 (65.6)** | **6 (1)** | | **19 (3.5)** | **524 (96.5)** | | **228 (39.8)** | **296 (60)** | | **20 (3.6)** | **539 (96.4)** | |

PCa: Prostate cancer

### Information provided by the physician regarding the PSA test

Out of the 573 respondents who stated that they had knowledge regarding the PSA test, 374 (65.4%) had received information from a physician regarding the PSA test (Table 3).

In a multivariable analysis (adjusted by previous prostate problems), respondents over 64 years and those aged 50–64 years were more likely to have received information (aOR 4.77, CI95% 2.92–7.79, and aOR 2.70, CI95% 1.74–4.20) than those aged 40–49 years (p<0.001) (not presented in table).

Out of the 374 respondents who reported being informed previously by a physician about the PSA test, 324 (86.6%) reported that the physician had informed them about the benefits of

**Table 3. Information provided by the physician regarding the PSA test.** (only those respondents who stated that they had knowledge regarding to the PSA test).

| Variables (n, %) | Total | Has a physician informed you about the PSA test? | | | Has a physician informed you about the advantages of the PSA test?* | | | Has a physician informed you about the disadvantages of the PSA test?* | | |
|---|---|---|---|---|---|---|---|---|---|---|
| | | No | Yes | P value | No | Yes | P value | No | Yes | P value |
| **Age (years)** | | | | <0.001 | | | 0.230 | | | 0.789 |
| 40–49 | 134 | 75 (56) | 59 (44) | | 12 (20.3) | 47 (79.7) | | 30 (50.8) | 29 (49.2) | |
| 50–64 | 249 | 82 (33.1) | 166 (66.9) | | 20 (12) | 146 (88) | | 80 (48.5) | 85 (51.5) | |
| >64 | 290 | 41 (21.6) | 149 (78.4) | | 18 (12.1) | 131 (87.9) | | 78 (52.3) | 71 (47.7) | |
| **Educational level** | | | | 0.139 | | | 0.004 | | | 0.002 |
| • Primary | 117 | 47 (40.5) | 69 (59.5) | | 16 (23.2) | 53 (76.8) | | 46 (66.7) | 23 (33.3) | |
| • Secondary/University | 446 | 148 (33.2) | 298 (66.8) | | 31 (10.4) | 267 (89.6) | | 136 (45.8) | 161 (54.2) | |
| **Health status** | | | | 0.054 | | | 0.281 | | | 0.585 |
| • Very bad/bad | 24 | 13 (54.2) | 11 (45.8) | | 0 | 11 (100) | | 7 (63.6) | 4 (36.4) | |
| • Normal | 241 | 74 (30.8) | 166 (69.2) | | 20 (12) | 146 (88) | | 85 (51.5) | 80 (48.5) | |
| • Very good/good | 308 | 111 (36) | 197 (64) | | 30 (15.2) | 167 (84.8) | | 96 (48.7) | 101 (51.3) | |
| **Previous prostate problems (self or relatives)** | | | | 0.003 | | | 0.453 | | | 0.010 |
| • No | 392 | 151 (38.5) | 241 (61.5) | | 34 (14.1) | 207 (85.9) | | 133 (55.2) | 108 (44.8) | |
| • Yes | 179 | 46 (25.8) | 132 (74.2) | | 15 (11.4) | 117 (88.6) | | 54 (41.2) | 77 (58.8) | |
| **Previous cancer (self or relatives)** | | | | 0.570 | | | 0.136 | | | 0.799 |
| • No | 324 | 109 (33.7) | 214 (66.3) | | 33 (15.4) | 181 (84.6) | | 108 (50.7) | 105 (49.3) | |
| • Yes | 247 | 89 (36) | 158 (64) | | 16 (10.1) | 142 (89.9) | | 78 (49.4) | 80 (50.6) | |
| **Total** | **573** | **198 (34.6)** | **374 (65.4)** | | **50 (13.4)** | **324 (86.6)** | | **189 (50.5)** | **185 (49.5)** | |

*Only those respondents who reported had received information related to PSA test.

the test. Respondents with secondary or university education (267, 89.6%) were more likely to receive information about the benefits of the test than those with primary education (53, 76.8%), p = 0.018. In contrast, less than 50% of the men (185, 49.5%) reported having received information about the risks associated with the PSA test. There were differences in the frequency of receiving information about the risks associated to the test according to education level and previous prostate problems (Table 3). In a multivariable analysis (adjusted by previous prostate problems), respondents with secondary or university studies were more likely to have received information about the risks related to the PSA test than those with primary education (aOR 2.56, CI95% 1.46–4.48) (p = 0.001) (not presented in table).

## Population practice in related to the PSA test

More than half of the respondents who stated they had knowledge regarding to the PSA test, had undergone a PSA test previously (432, 66.1%), with 80% of them having done so in the two years preceding the study (mainly in respondents over 50 years old). Respondents over 64 years (174, 92.1%) and those aged 50–64 years (202, 81.8%) were more likely to have had a PSA test than those aged 40–49 years (56, 42.4%), p<0.001 (Table 4). Furthermore, 383 (88.6%) respondents expressed their willingness to receive the PSA test again. Out of the 136 respondents who had not had the test, 105 (77.2%) expressed their willingness to undergo the test if a physician recommended it (details not presented in tables).

The recommendation of their physician, family or friends was one of the main reasons for PSA testing among the 432 respondents who had previously undergone the test (169, 41.85%). A relevant percentage of respondents (95, 23.5%) had been unaware that the test

**Table 4. Description of the respondents' practice regarding the PSA test.** (only those respondents who stated that they had knowledge regarding to the PSA test).

| Variables (n, %) | Total | Have you had a PSA test? | | | When did you last have a PSA test?* | | | Reasons to have a PSA test* | | | |
|---|---|---|---|---|---|---|---|---|---|---|---|
| | | No | Yes | p value | <2 years | >= 2 years | p value | Family/friends/physician recommendations | Unaware that a request had been made | Others* | p value |
| **Age (years)** | | | | <0.001 | | | 0.047 | | | | <0.001 |
| 40–49 | 134 | 76 (57.6) | 56 (42.4) | | 38 (67.9) | 18 (32.1) | | 28 (50.9) | 19 (34.5) | 8 (14.5) | |
| 50–64 | 249 | 45 (18.2) | 202 (81.8) | | 162 (81) | 38 (19) | | 66 (34.9) | 53 (28) | 70 (37) | |
| >64 | 290 | 15 (7.9) | 174 (92.1) | | 144 (82.8) | 30 (17.2) | | 75 (46.9) | 23 (14.4) | 62 (38.8) | |
| **Educational level** | | | | 0.155 | | | 0.424 | | | | 0.783 |
| • Primary | 117 | 22 (19.1) | 93 (80.9) | | 72 (77.4) | 21 (22.6) | | 34 (39.1) | 22 (25.3) | 31 (35.6) | |
| • Secondary/University | 446 | 113 (25.5) | 330 (74.5) | | 267 (81.2) | 62 (18.8) | | 133 (43) | 70 (22.7) | 106 (34.3) | |
| **Health status** | | | | 0.402 | | | 0.232 | | | | 0.597 |
| • Very bad/bad | 24 | 6 (25) | 18 (75) | | 12 (66.7) | 6 (33.3) | | 9 (52.9) | 5 (29.4) | 3 (17.6) | |
| • Normal | 241 | 50 (21.1) | 187 (78.9) | | 147 (78.6) | 40 (21.4) | | 75 (42.6) | 42 (23.9) | 59 (33.5) | |
| • Very good/good | 308 | 80 (26.1) | 227 (73.9) | | 185 (82.2) | 40 (17.8) | | 85 (40.3) | 48 (22.7) | 78 (37) | |
| **Previous prostate problems (self or relatives)** | | | | 0.004 | | | 0.807 | | | | 0.002 |
| • No | 392 | 107 (27.5) | 282 (72.5) | | 226 (80.7) | 54 (19.3) | | 102 (38.2) | 56 (21) | 109 (40.8) | |
| • Yes | 179 | 29 (16.4) | 148 (83.6) | | 118 (79.7) | 30 (20.3) | | 66 (48.5) | 39 (28.7) | 31 (22.8) | |
| **Previous cancer (self or relatives)** | | | | 0.204 | | | 0.987 | | | | 0.273 |
| • No | 324 | 84 (26) | 239 (74) | | 190 (80.2) | 47 (19.8) | | 86 (38.1) | 57 (25.2) | 83 (36.7) | |
| • Yes | 247 | 52 (21.4) | 191 (78.6) | | 153 (80.1) | 38 (19.9) | | 81 (46) | 38 (21.6) | 57 (32.4) | |
| **TOTAL** | **573** | **136 (23.9)** | **432 (66.1)** | | **442 (80.2)** | **106 (19.8)** | | **169 (41.8)** | **95 (23.5)** | **140 (34.7)** | |

*Only those respondents who have had a PSA test.

had been requested when it was requested. This was more common among those aged 40–49 years (19, 34.5%) and those ranged 50–64 years (53, 28%) compared to those over 64 years (23, 14.4%). Respondents aged 50–64 years (70, 37%) and those over 64 years (62, 38.8%) were more likely to have other reasons (such as believing they were at risk of PCa or having prostate symptoms) for having the PSA test, in comparison with men aged 40–49 years (8, 14.5%), p<0.001.

Of the 432 respondents who had undergone a PSA test previously, 348 (80.5%) had received information from a physician about the test; in contrast, of the 136 respondents who had not had previously a PSA test, 25 (18.4%) had received information from physician about the test.

## Discussion

The results of this survey indicate that a high percentage of respondents report having knowledge regarding to PCa, mainly those with higher education. However, only half of the interviewed men reported having knowledge about the PSA test, mainly those older than 50 years of age, and those with higher education. However, regardless of education, frequent misconceptions about opportunistic PCa screening with PSA were also observed, mainly in younger respondents and in those with primary education.

A relevant percentage of respondents stated that they had been informed by a physician about the PSA test, mainly in patients over 50 years; however, patients with secondary or university education were more likely to have received information about benefits and risks of the tst in comparison with respondents with primary education. Mores than 65% of respondents, mainly over 50 years of age, had a PSA test, with 80% of them having done so in the last two years. Among men aged 40–49 years old, receiving information from their physician, family or friends was one of the main reasons for PSA testing. Older patients in contrast, tended to have the test mainly because they thought they were at risk of PCa or they had prostate symptoms.

Previous studies have suggested that men who are more knowledgeable about PCa are more likely to be screened [8]. In our study, nearly 95% of respondents reported having knowledge regarding to PCa, a higher percentage than those shown in a survey carried out in Italy, in which 82.1% reported having heard of PCa before. Other studies, mainly carried out in African countries, showed lower levels: in South Africa (45.7%) [17] and in Uganda (54.1%) [18]. In our study, younger respondents were more likely to express concern regarding having PCa and to think that men younger 50 years were at higher risk of developing PCa.

There are significant geographic differences between different regions regarding percentage of awareness of the PSA test. In our study, 67.7% of the respondents reported having knowledge about the PSA test. This percentage is similar to other European countries. In a previous Italian survey for instance, 72.7% of the respondents reported having knowledge about the PSA test [13]. In contrast, this value is lower in Asia and in Africa. A Chinese survey in general population (including males and females) found that 37.9% of the respondents were aware of the PSA test [19]. These percentages are similar to another Asian study, in which although 96.5% of the 600 Korean respondents knew about PCa only 9.7% of men over the age of 40 recognized the value of PSA as a screening test [20]. The percentage of awareness of the PSA test is even lower in Africa, for instance, in South Africa where less than a quarter of respondents reported having prior knowledge about the PSA test [21] and in Nigeria, only 25.1% had heard about the PSA test [22]. These differences might be due to the significant variability in health care facilities in the regions.

As in other studies [9, 23], men over 50 years and those with secondary or university studies were more likely to report having knowledge regarding to the PSA test. There was, however, significant misconceptions about screening, mainly among younger men, which may influence their decisions. For instance, one every three of respondents believed that the appropriate age for PSA testing was under 50 years old (particularly respondents aged 40–49 years old, in line with their concern regarding having PCa). Moreover, 60% of respondents believed that the PSA test decreased the risk of developing the disease (mainly those individuals with primary education).

In our study, individuals with higher educational level were more likely to receive information about the PSA test from clinicians, but having a higher educational level was not related to a higher probability of PSA testing. Scherer et al. [24] reported that individuals with higher levels of education tend to prefer less health care. In Australia, in contrast, higher educated men with private health insurance were more likely to receive PSA tests than other

demographic groups [23]. In Spain, nearly 95% of population is covered by public health care. In this study, we did not include data from private clinics and thus, we could not assess if these differences existed.

Previous studies showed that patient–provider communication about the pros and cons of PSA testing continues to be deficient. In a previous study that included patients with PCa, only half of the men received this information and a third received no information prior to testing [17]. Another study in USA showed that health care providers emphasized the pros of testing in 71.4% of discussions but infrequently addressed the cons (32.0%) [25]. In our study, although 65.4% of respondents received information about the benefits of the test, only 49.5% were informed about the associated risks. Subjects had a higher educational level were more likely to receive information related to both benefits and risks.

Two out if three respondents from our study had undergone a PSA test, a percentage higher than in studies in other countries. In a previous Italian survey in 2017, only 29.6% had received a PSA test, which was similar to another study conducted in South Africa in 2015 in which only 28.3% of men had received a PSA test [17]. A survey in Portugal [26] carried out in 2020 showed that 44.2% of men had been submitted to PCa screening; 13.8% received only DRE, 12.2% received only PSA test, and 18.2% received both DRE and PSA tests. There was no association between factors such as health status and testing frequency, which was in line with other research indicating that individuals who asked for more healthcare did not necessarily have more health problems [27]. In fact, in our study, individuals in a bad or very bad state of health were less likely to harbor concerns about PCa in comparison with men in a normal, good or very good state of health.

Nearly 80% of respondents who had had a PSA test, had received information about the PSA test from a physician, and one of the primary motivations for undergoing a PSA test, cited by 41.8% of the patients who had undergone the test, was recommended from a physician, family or friend. This percentage was lower to than figures reported in previous studies, in which the percentage was nearly 50% [13, 26]. Over 77% of surveyed men who had not undergone the PSA test expressed their willingness to undergo it if recommended to do so by their clinicians. Hence, the role of clinicians continues to be crucial in influencing patient choices.

## Implications for clinical practice

Although several biomarkers and risk algorithms have been developed to support clinicians in PCa screening, the PSA blood test remains the first-line screening test of choice according to previous randomized clinical trials [28, 29] and is widely used in clinical practice [23]. Hence, the results of this study may be useful in the development of strategies to implement tailored SDM strategies.

There is clearly a need to enhance the procedures for informing patients, particularly surrounding the potential risks of the test, to younger men and to those with lower levels of education. The inclusion of decision aids could serve as valuable support, as demonstrated in previous studies [30]. Enhancing the understanding of how individual factors correlate with diverse healthcare approaches may also contribute to more effective communication.

## Limitations

Our study has certain limitations. Firstly, it was conducted in a specific Spanish region, and caution should be exercised in generalizing the findings to other settings. However, the inclusion of a randomly selected general population suggests that the results may have similarities with other Spanish population. Since the survey was conducted via telephone, there could be a

selection bias; however, 80% of the selected individuals participated in the interview without differences by age or municipality. The inclusion of a high percentage of individuals with higher educational levels could reflect such bias, as according to data from the Valencian Community, 62% of the male population has higher education levels (slightly lower than shown in this study). Nonetheless, the differences reflected between educational level and individuals' knowledge and practices are similar to those seen in previous studies. The cross-sectional design limits the establishment of a clear association between dependent and independent variables. Nonetheless, the collection of numerous relevant variables and adjustment for confounders enabled us to conduct a thorough analysis of the situation.

On the positive side, this survey presents advantages over previous studies. The questionnaire underwent a Delphi process with experts, and both content validity and reliability were evaluated beforehand. Furthermore, it is the first survey on PSA knowledge and practices conducted since the updating of available guidelines from the U.S. Preventive Services Task Force [5] and the European Association of Urology [6].

## Conclusions

This survey represents the first study, to the best of our knowledge, which investigates the correlations between personal factors and individuals' knowledge and practices. The differences found in the knowledge of PCa and its risks, as well as in the understanding of the PSA test and its characteristics according to age and educational level, highlight the need to improve the information provided by clinicians and tailor it to the characteristics of the patients. The prevalence of PSA testing in our population was higher than in similar studies, and men showed a willingness to undergo retesting, mainly upon physician recommendation. The relevant percentage of misconceptions about PCa screening underscores the imperative to incorporate additional procedures for patient education to ensure that decision making regarding PSA testing is truly shared.

## Supporting information

**S1 Table. STROBE statement—Checklist of items that should be included in reports of cross-sectional studies.**
(DOC)

## Acknowledgments

Investrategia, for the support in the performance of the survey. Jessica Gorling for the English editing.

## Author Contributions

**Conceptualization:** Lucy A. Parker, Ildefonso Hernandez-Aguado, Blanca Lumbreras.

**Formal analysis:** Lucy A. Parker, Juan-Pablo Caballero-Romeu, Elisa Chilet-Rosell.

**Funding acquisition:** Blanca Lumbreras.

**Methodology:** Lucy A. Parker, Juan-Pablo Caballero-Romeu, Elisa Chilet-Rosell, Ildefonso Hernandez-Aguado, Luis Gómez-Pérez, Pablo Alonso-Coello, Ana Cebrián, Maite López-Garrigós, Irene Moral-Pélaez, Elena Ronda, Mercedes Gilabert, Carlos Canelo-Aybar, Ignacio Párraga-Martínez, Mª del Campo-Giménez.

**Resources:** Blanca Lumbreras.

**Supervision:** Blanca Lumbreras.

**Writing – original draft:** Lucy A. Parker, Elisa Chilet-Rosell, Blanca Lumbreras.

**Writing – review & editing:** Juan-Pablo Caballero-Romeu, Ildefonso Hernandez-Aguado, Luis Gómez-Pérez, Pablo Alonso-Coello, Ana Cebrián, Maite López-Garrigós, Irene Moral-Pélaez, Elena Ronda, Mercedes Gilabert, Carlos Canelo-Aybar, Ignacio Párraga-Martínez, Mª del Campo-Giménez, Blanca Lumbreras.

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
