## [Decision Letter · Decision Letter 0]

22 Mar 2024

PONE-D-24-07506Knowledge and practices regarding prostate cancer screening in Spanish men: the importance of personal and clinical characteristics (PROSHADE study)PLOS ONE

Dear Dr. Lumbreras,

Thank you for submitting your manuscript to PLOS ONE. After careful consideration, we feel that it has merit but does not fully meet PLOS ONE’s publication criteria as it currently stands. Therefore, we invite you to submit a revised version of the manuscript that addresses the points raised during the review process.

We look forward to receiving your revised manuscript.

Kind regards,

Ismaheel Lawal, MD, PhD

Academic Editor

PLOS ONE

Journal Requirements:

" Instituto de Salud Carlos III, code PI20/01334"

Reviewers' comments:

Reviewer's Responses to Questions

**Comments to the Author**

1. Is the manuscript technically sound, and do the data support the conclusions?

Reviewer #1: Yes

Reviewer #2: Yes

2. Has the statistical analysis been performed appropriately and rigorously? 

Reviewer #1: Yes

Reviewer #2: Yes

3. Have the authors made all data underlying the findings in their manuscript fully available?

Reviewer #1: Yes

Reviewer #2: Yes

4. Is the manuscript presented in an intelligible fashion and written in standard English?

Reviewer #1: Yes

Reviewer #2: Yes

5. Review Comments to the Author

Reviewer #1: In the modern era of prostate cancer detection, the most important thing is how regularly men are exposed to PSA testing, and since most prostate cancers are detected at a localized or locally advanced stage without symptoms, the systems in place to make PSA testing universal, and the availability of this information on a regular basis, are very different in different countries. Therefore, globally, the results of a cross-sectional study from Spain, a developed country in Western Europe, that gives us an indication of the universal situation in 2022, are very relevant. As the authors note, the 2012 USPSFT guideline against widespread PSA testing has served to replicate US standards in other countries, without taking into account the different circumstances in each country.

Minor point

1. of the 95.4% who said they knew about prostate cancer, the level of knowledge can vary widely. I would like to see more discussion of the very different 61.9% of people who said they knew about the PSA, and the contrast between awareness of PCa and awareness of the PSA in the abstract, as it would be rather odd not to have heard of prostate cancer itself through various media if a binary question was attempted. I think this difference in awareness of these two questions is the most important point to emphasize in the paper, as it is the only preparation for prostate cancer that the public can actually do.

2. Awareness of PSA is very low, especially in Asia. In South Korea, which is considered a developed country, only 9.7% of men over the age of 40 recognized the value of PSA as a screening test in a random survey of men over the age of 40. Please cite additional country-specific data on the social awareness of PSA testing in general in Europe, Africa, as well as Asia.

Jong Hyun Pyun, Seok Ho Kang, Ji Youn Kim, Jae Eun Shin, In Gab Jeong, Jong Wook Kim, Tae Il No, Jong Jin Oh, Ji Hyung Yu, Ho Seok Chung, Seong Soo Jeon Survey Results on the Perception of Prostate-Specific Antigen and Prostate Cancer Screening Among the General Public Korean J Urol Oncol. 2020;18(1):40-46. Published online April 29, 2020 ( https://doi.org/10.22465/kjuo.2020.18.1.40 )

Reviewer #2: This is a well-written and methodologically sound paper meriting publication. It deals with the controversial field of prostate cancer screening and knowledge determinants for PSA screening to ultimately inform their screening decision aid for Spanish men

For the introduction, line 104-105, I suggest making this statement specific -there is limited data published on knowledge of PSA and of screening tests in general and the recent published data is mainly coming for disadvantaged and high-risk populations - see latest references the authors seem to have missed -doi: 10.1155/2019/2463048 , 10.4314/ejhs.v33i3.19 , https://doi.org/10.1177/1557988316689497

For results section line 302-303 -do the authors mean with respect to sociodemographic characteristics?

Line 496 I suggest replacing primary studies with primary education

Line 581 -surely it should be similarities with other Spanish populations?

6. PLOS authors have the option to publish the peer review history of their article (what does this mean?). If published, this will include your full peer review and any attached files.

Reviewer #1: **Yes: **Young Hwii Ko

Reviewer #2: **Yes: **Dr Maureen Joffe

---

## [Author Response · Author response to Decision Letter 0]

18 Apr 2024

Thank you for your thoughtful comments on our manuscript “Knowledge and practices regarding prostate cancer screening in Spanish men: the importance of personal and clinical characteristics (PROSHADE study)”.

Below, we answer the reviewer’s questions.

The revised version is attached.

Neither the manuscript that we submit to you nor any similar paper, in whole or in part, has been or will be submitted to or published in any other scientific journal.

All authors accept the uniform requirements for biomedical journals, as well as those required by PLos One. All authors have read and approved the paper and will be happy to enclose signatures if required.

Research was funded by the Instituto de Salud Carlos III, code PI20/01334, co-financed with FEDER funds from the European Union “A way of doing Europe”. The funders had no role in study design, data collection and analysis, decision to publish, or preparation of the manuscript.

We thank you for your consideration and look forward to hearing from you.

Blanca Lumbreras

Department of Public Health, History of Science and Gynecology, Miguel Hernández University , and CIBER en Epidemiología y Salud Pública, Spain.

Crtra Alicante-Valencia km 81. Sant Joan d’Alacant, 03550, Alicante, Spain

Phone number: +34965919510

Email: blumbreras@umh.es

Journal Requirements:

Done

" Instituto de Salud Carlos III, code PI20/01334"

We have included this statement in the cover letter and in the manuscript.

We have moved the ethic statement to the method section.

We have included captions of the Supporting Information files at the end of the manuscript (S1 table STROBE Statement—Checklist of items that should be included in reports of cross-sectional studies).

Done 

Reviewers' comments:

Reviewer #1: In the modern era of prostate cancer detection, the most important thing is how regularly men are exposed to PSA testing, and since most prostate cancers are detected at a localized or locally advanced stage without symptoms, the systems in place to make PSA testing universal, and the availability of this information on a regular basis, are very different in different countries. Therefore, globally, the results of a cross-sectional study from Spain, a developed country in Western Europe, that gives us an indication of the universal situation in 2022, are very relevant. As the authors note, the 2012 USPSFT guideline against widespread PSA testing has served to replicate US standards in other countries, without taking into account the different circumstances in each country.

Minor point

1. of the 95.4% who said they knew about prostate cancer, the level of knowledge can vary widely. I would like to see more discussion of the very different 61.9% of people who said they knew about the PSA, and the contrast between awareness of PCa and awareness of the PSA in the abstract, as it would be rather odd not to have heard of prostate cancer itself through various media if a binary question was attempted. I think this difference in awareness of these two questions is the most important point to emphasize in the paper, as it is the only preparation for prostate cancer that the public can actually do.

The reviewer is right, most of the interviewed men had knowledge about PCa (95.4%). Of the 1,067 surveyed men, only 573 (50.3%) reported that they had knowledge about the PSA test (67.7% of the respondents, we made a mistake in the initial data). 

As the reviewer pointed out, this is very relevant since the lack of awareness related to the PSA test is a barrier to the shared decision making in PCa screening.

We have included this sentence in the abstract section (page 3, lines 64-65 and lines 71-72).

2. Awareness of PSA is very low, especially in Asia. In South Korea, which is considered a developed country, only 9.7% of men over the age of 40 recognized the value of PSA as a screening test in a random survey of men over the age of 40. Please cite additional country-specific data on the social awareness of PSA testing in general in Europe, Africa, as well as Asia.

Jong Hyun Pyun, Seok Ho Kang, Ji Youn Kim, Jae Eun Shin, In Gab Jeong, Jong Wook Kim, Tae Il No, Jong Jin Oh, Ji Hyung Yu, Ho Seok Chung, Seong Soo Jeon Survey Results on the Perception of Prostate-Specific Antigen and Prostate Cancer Screening Among the General Public Korean J Urol Oncol. 2020;18(1):40-46. Published online April 29, 2020 ( https://doi.org/10.22465/kjuo.2020.18.1.40 )

We have included a more detailed description about the difference awareness of PSA test and PCa in different regions around the world (discussion, pages 27-28, lines 560-579).

Reviewer #2: This is a well-written and methodologically sound paper meriting publication. It deals with the controversial field of prostate cancer screening and knowledge determinants for PSA screening to ultimately inform their screening decision aid for Spanish men

For the introduction, line 104-105, I suggest making this statement specific -there is limited data published on knowledge of PSA and of screening tests in general and the recent published data is mainly coming for disadvantaged and high-risk populations - see latest references the authors seem to have missed -doi: 10.1155/2019/2463048 , 10.4314/ejhs.v33i3.19 ,https://doi.org/10.1177/1557988316689497

Following the reviewer’s suggestion, we have highlighted this point and we have introduced these relevant references (introduction, page 6, lines 132-134).

For results section line 302-303 -do the authors mean with respect to sociodemographic characteristics?

Yes, we referred to sociodemographic characteristics. We have added this concept to clarify the sentence (results, page 15, line 347).

Line 496 I suggest replacing primary studies with primary education

Done (discussion, page 26, line 543)

Line 581 -surely it should be similarities with other Spanish populations?

Yes, we have added this term (discussion, page 29, line 654).

---

## [Editor Report · Decision Letter 1]

22 Apr 2024

Knowledge and practices regarding prostate cancer screening in Spanish men: the importance of personal and clinical characteristics (PROSHADE study)

PONE-D-24-07506R1

Dear Dr. Lumbreras,

We’re pleased to inform you that your manuscript has been judged scientifically suitable for publication and will be formally accepted for publication once it meets all outstanding technical requirements.

Kind regards,

Ismaheel Lawal, MD,

Academic Editor

PLOS ONE
---

## [Editor Report · Acceptance letter]

8 May 2024

PONE-D-24-07506R1 

PLOS ONE

Dear Dr. Lumbreras, 

I'm pleased to inform you that your manuscript has been deemed suitable for publication in PLOS ONE. Congratulations! Your manuscript is now being handed over to our production team.

Kind regards, 

on behalf of

Dr. Ismaheel Lawal 

Academic Editor

PLOS ONE